# Anticipatory behaviour as an indicator of the welfare of dairy calves in different housing environments

**Heather W. Neave**[1,2], **James R. Webster**[1], **Gosia Zobel**[1] *

**1** Animal Behaviour and Welfare, AgResearch Ltd., Ruakura Research Centre, Hamilton, New Zealand,
**2** Animal Welfare Program, Faculty of Land and Food Systems, University of British Columbia, Vancouver, B. C., Canada

* gosia.zobel@agresearch.co.nz

## Abstract

Anticipatory behaviour occurs in the period before a reward or other positive event is presented and has been interpreted as an indicator of the welfare and emotional state of animals. The use of this indicator has received limited attention in dairy calves. Therefore, we investigated how anticipatory behaviour is affected by housing environment and reward quality, and if anticipatory behaviour changes when reward quality changes unexpectedly. Sixteen pairs of calves were assigned to treatments in a 2 x 2 factorial design (two housing environment and two reward quality combinations). Housing was either basic (2 m²/calf, river stone surface) or enriched (5 m²/calf, woodchip, and enrichment items), and the reward was access to either an additional basic or enriched pen. Calves were conditioned to anticipate reward pen access; anticipatory behaviour toward receiving the reward pen was measured. Signaling reward access increased the frequency of transitions between behaviours and duration of touching and looking at the signal and exit door. Basic-housed calves showed more anticipatory behaviour (increased frequency of transitions between behaviours) and decreased latency to access the reward compared to enriched-housed calves, but the reward pen quality had no effect on anticipatory behaviour. When the reward pen quality changed from enriched to basic unexpectedly, resulting in sudden reward loss, basic-housed calves decreased, while enriched-housed calves increased, anticipatory behaviour. However, there was no change in anticipatory behaviour during reward gain (change from basic to enriched reward pen). Our findings align with previous work showing that animals in basic housing show more anticipation for a reward, and demonstrate suppressed behavioural response when experiencing reward loss, suggesting greater sensitivity to reward. Sensitivity to reward has associations with mood state; thus, calves in basic environments may experience a more negative emotional state, and thus reduced welfare, compared to calves in enriched environments.

**Data Availability Statement:** All data generated and analysed in this study are available in the Mendeley repository at: https://data.mendeley.com/datasets/zkvdxbpjdf/draft?a=9026bb73-86e5-4398-88cb-0d11c31ba3de.

**Funding:** The study funding was provided by AgResearch Core funding (A19041). At the time of study completion, author HN was supported by the Natural Science and Engineering Research Council Canada Graduate Scholarship and the Michael Smith Foreign Study Supplement. Furthermore, the study funders provided support in the form of salaries for author GZ but did not have any additional role in the study design, data collection and analysis, decision to publish, or preparation of the manuscript. The specific roles of the authors are articulated in the 'author contributions' section.

**Competing interests:** The commercial funding received does not alter our adherence to PLOS ONE policies on sharing data and materials. The commercial affiliation of GZ and HN with AgResearch Ltd. (a New Zealand crown research institution), does not alter our adherence to PLOS ONE policies on sharing data and materials.

## Introduction

The opportunity to experience positive emotional states is one of the primary tenets of good animal welfare [1], and some argue that it is the most important consideration when evaluating whether an animal has 'a good life' [2]. Much of the recent animal emotion research has measured cognitive processes–appraisal of stimuli or events–to identify states affecting how an animal perceives its world [3]. One approach is to use anticipatory behaviour to identify an animal's emotional state through goal-directed behaviour [4, 5]. This approach has received limited attention in farm animals, especially dairy cattle.

Anticipatory behaviour is displayed in the period between the signalling of an upcoming event (conditioned stimulus) and the presentation of the event (unconditioned stimulus, where the event is typically a positive event or reward). This behaviour reflects a motivational state and provides information about how the event is perceived [6]. In the case of positive events, anticipatory behaviour arises as a result of the release of endorphins and dopamine before the actual arrival of the reward [6]. Thus, anticipatory behaviour may itself be experienced as positive, and is mediated by the mesolimbic dopaminergic system, providing information on the reward and motivational 'centers' of the animal [6, 7]. The release of dopamine and endorphins promotes locomotory activity [8], so anticipatory behaviour is often measured as increased frequency of behaviours (i.e., transitions between different behaviours, reported in lambs [9], horses [10] and mink [11]), increased activity (e.g., pigs [12]) or behaviours directed at the conditioned stimulus signalling arrival of the reward (e.g., chickens [13]).

The link between anticipatory behaviour and the dopaminergic system means that observation of anticipatory behaviour can indicate an animal's sensitivity to reward. This sensitivity has important associations with emotional states such as depression and anxiety (e.g. [14, 15] and is affected by previous experiences, such as an animal's housing environment. For instance, rats housed in conventional cages showed more anticipatory behaviours for a food reward than those in enriched cages [16, 17], and isolated rats anticipated food rewards and social contact more than group-housed rats [18]. These studies suggest that sensitivity to reward increases when animals are under stressful conditions involving deprivation of stimuli, likely due to a greater value that is placed on the reward by the deprived individual [6]. Thus, anticipatory behaviour has been used to identify an animal's affective state, and also to make inferences regarding its welfare (see reviews by Watters [5] and Anderson et al. [19]).

There is some evidence that the level of anticipatory behaviour (i.e. frequency of behaviours) can indicate the value of the reward to the animal, such as environmental enrichment. For example, rats showed greater anticipatory behaviour before transfer from a standard home cage to an enriched cage than when transferred to another standard cage [20], laying hens exhibited more anticipatory behaviour for dustbathing than a food reward [21], lambs showed similar levels of anticipatory behaviour before a food reward or the opportunity to play [9], and dolphins showed more anticipatory behaviour for interaction with human trainers than for a toy [22]. The emotional states of animals in different housing environments can also be inferred by their response to a change in reward value. For instance, rats and pigs ran slower to reach a reward that unexpectedly decreased in reward size, and this effect was more pronounced in standard or barren housed animals [14, 23]. It is thought that the discrepancy between the expected and actual level or type of reward induces a 'disappointment-like' aversive affective state, and animals responding more negatively to this discrepancy in reward (such as the rats and pigs in barren housing) can be an indication of the animal's negative background affective state [24].

Dairy calves are a good model for studying anticipatory behaviour as an indicator of emotional state when in different housing environments; in North America, calves are typically

housed in isolation in a physically limited space until weaning at around 8 weeks of age (see review by Costa et al. [25]) and the expression of natural behaviours is limited [26], leading to poor welfare. In New Zealand, dairy calves are permitted social contact during the pre-weaning period, but early life housing can still restrict natural and pleasurable behaviours (e.g. grooming), and some bedding options reduce lying and playing (e.g. river stones [27, 28]). To our knowledge, anticipatory behaviour has not been used to investigate the emotional states of dairy calves in different housing environments (such as basic or enriched pens) or how they value rewards. However, this work is essential to understanding if improvements in calf housing may be rewarding.

The objectives of this study were to investigate how anticipatory behaviour of dairy calves is affected by housing environment and reward quality, and whether anticipatory behaviour is affected when reward quality changes. We predicted that calves housed in a basic pen would show greater anticipatory behaviours for access to a reward pen compared to calves housed in an enriched pen, especially if the reward pen was enriched. We also predicted that calves housed in a basic pen would show a greater change in anticipatory behaviour when the reward pen quality changed, especially from enriched to basic. Together these results would support our hypothesis that calves experience reduced welfare in basic pens.

## Materials and methods

This study was conducted from March to May 2016 at the Tokanui Dairy Research Facility, Te Awamutu, Waikato, New Zealand. All procedures performed were approved by the Ruakura Animal Ethics Committee in Hamilton, New Zealand (#13822) and by the University of British Columbia Animal Care Committee (#A16-0213) in Vancouver, BC, Canada.

### Animal management and treatment groups

A total of 64 male and female dairy calves (Holstein-Friesian cross) were leased from a commercial farm and enrolled upon arrival at the research facility. The study was performed in two replicates of 32 calves with each replicate lasting 22 days (Rep 1: n = 15 male, n = 17 female; Rep 2: n = 12 male, n = 20 female). Calves were (mean ± SD) 9.2 ± 4.6 d of age and weighed 50.5 ± 7.0 kg at enrollment. Calves were randomly assigned in pairs to one of four treatments in a 2 x 2 factorial arrangement (n = 16 each): one of two home pen housing treatments, and one of two reward pen treatments to which calves had set daily access. Home pen treatments were either (1) 'basic' housing, consisting of 2 $m^2$ per calf and bedded with river stones (used by some New Zealand farmers, [28]), or (2) 'enriched' housing, consisting of 5 $m^2$ per calf, bedded with woodchip and containing 3 'enrichment' items: a wall-mounted automated brush (mini swinging brush MSB, DeLaval, Tumba, Sweden), a 1 m long manila rope attached to the pen wall opposite the brush (Manila rope 28 mm 3 strand, Action Outdoors Ltd, Auckland, NZ) and a small pile of straw. Calf use of these enrichment items has been described by Zobel et al. [29]. Home pens contained water and concentrate feeders mounted at the back of the pen, and a hay feeder mounted at the front of the pen. Reward pens were identical to the basic (B) or enriched (E) home pens, except without feeders. Thus, calves experienced either a similar or different 'reward pen' environment to their home pen. Treatments were balanced for calf weight and sex, and all pairs included one heifer and one bull calf, with the exception of 4 pairs of heifer calves that were balanced across treatments. All heifer calves were colour marked on their back, and in the case of two heifers in a pair, one was randomly selected for colour marking (Tell Tail paint, FIL NZ Ltd, Mount Maunganui, New Zealand).

Calves were fed 8 L/d of whole milk from a nipple (divided into two 4 L meals fed twice daily at 0700 and 1500 h). Each milk meal also contained 12.5 g of beta-cyclodextrin (Exagen;

Professional Veterinary Distributors Ltd, Auckland, New Zealand). In each home pen, calves had access to *ad libitum* hay, calf starter concentrate (Gusto Calf Blend 16, Allied Grain Co-operative Ltd., TeAwamutu, New Zealand) and water from wall-mounted troughs that were replenished daily. Straw in the enriched pen was replenished every second day. Basic home pens were cleaned twice per week by washing the river stones with water. Enriched home pens had woodchip bedding added once per week. Fly spray (RipCordⓇ, BASF, Auckland, NZ; approximately 5 mL/pen) was applied to the pen walls and coat of each calf following the conditioning or testing procedures every other day during Rep 1, and every day during Rep 2.

## Conditioning procedure (training)

Following one day of habituation to their home pen environment, all calves were subjected to a classical conditioning paradigm outlined in Fig 1A. A yellow flashing light mounted on the side wall next to door of the home pen was repeatedly paired with the opening of the exit door of the home pen. When this door was open, calves could access their assigned reward pen by walking along a woodchip bedded alleyway (2.5 m long with black plastic tarpaulin walls). The interval of time between the light signal and the home pen door opening was gradually increased over a period of 12 days (conditioning phase; Fig 1B) to a maximum of 60 s. On Day 1, the light signal was switched on and the home pen exit door was immediately opened. The light was turned off after the door was fastened open, and calves were allowed 30 min to explore the alleyway and reward pen. Calves were not forced to enter the reward pen. This was repeated on Day 2, but if one or both calves had not entered the reward pen after 15 min, the experimenter gently pushed the calves into the reward pen after which the calves could leave if they chose. This procedure was repeated from Day 2 to 12 of the conditioning procedure. On Day 3, the interval between the light and the door opening was increased to 1 s, and from Day 4 to 9 the interval increased by 5 s each day. From Day 10 to 12 the interval increased by 10 s each day to the maximum interval of 1 min on Day 12. On Day 4 to 12, to ensure calves saw and had an opportunity to respond to the light, their attention was gained by the experimenter

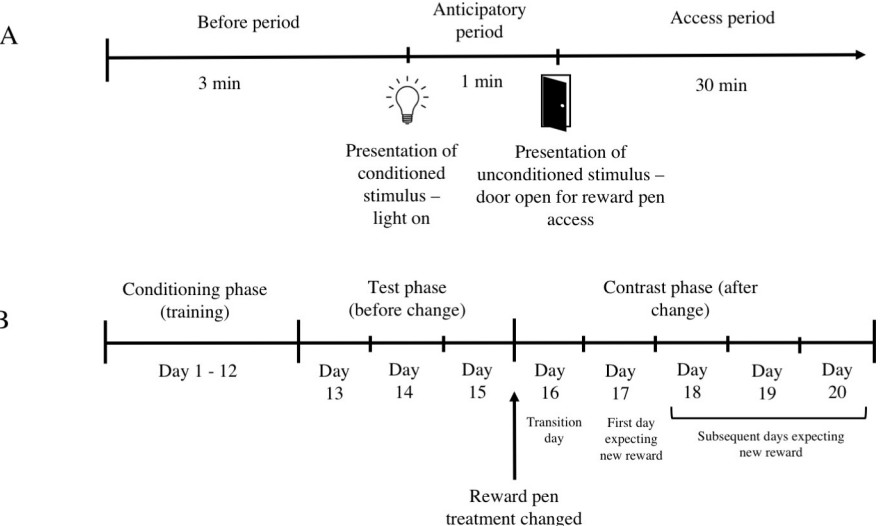

**Fig 1. Overview of experimental design.** (A) Each day of the test and contrast phases consisted of 3 periods during which behaviours of the calves were observed: before (3 min), anticipatory (1 min) and access (30 min) periods. (B) Description of days within each phase (conditioning, test and contrast phases) and each stage of change (before change and after change).

slowly waving a white flag over the top of pen wall at the time the light was switched on. The flag was waved for 5 s and then removed for the remainder of the anticipatory period. The calves were unable to see the experimenter or other calves at any point during the conditioning procedure (the walls of the pen were 2.4 m high and thus prevented viewing outside the pen).

All calves were subjected to the conditioning procedures each day from 0800 to 1200 h. Four pairs of calves of the same treatment (i.e., B-B, B-E, E-B or E-E) were conditioned simultaneously, and each pair had exclusive access to their reward pen during the 30 min of access. This process was repeated until all 16 pairs of calves (4 on each treatment) completed the conditioning procedure for the day. Order of conditioning of pairs rotated between and within treatments. Alleyways were attached between the home and reward pens each morning for the first 8 pairs of calves, and once conditioning for these pairs was completed, alleyways were moved to connect the home and reward pens of the second set of 8 pairs (see Fig 2 for an overview of this layout). There was a 15 min wait period after alleyway set up before the conditioning procedure began, to minimize the potential of this acting as a conditioned stimulus signaling access to the reward pen. After each 30 min of reward pen access, reward pens were cleared of any feces, and straw in the enriched pen was collected back into a small pile; straw was replaced at the end of each day. The same conditioning procedure was repeated for Rep 2.

## Testing and contrast phase procedures

Following the conditioning phase, calves began the test phase which consisted of 3 days (day 13 to 15; Fig 1B). Calves were tested in the conditioning procedure with the 60 s interval between the light and flag signal and the exit door opening to permit access to the reward pen. Calves were not pushed into the reward pen if they chose not to enter. An experimenter sitting next to the reward pen, but out of sight, recorded vocalizations during the 30 min *access* period.

After completing the test phase, calves began the contrast phase which consisted of 5 days (day 16 to 20; Fig 1B). At completion of day 15, all reward pens were cleaned out and changed

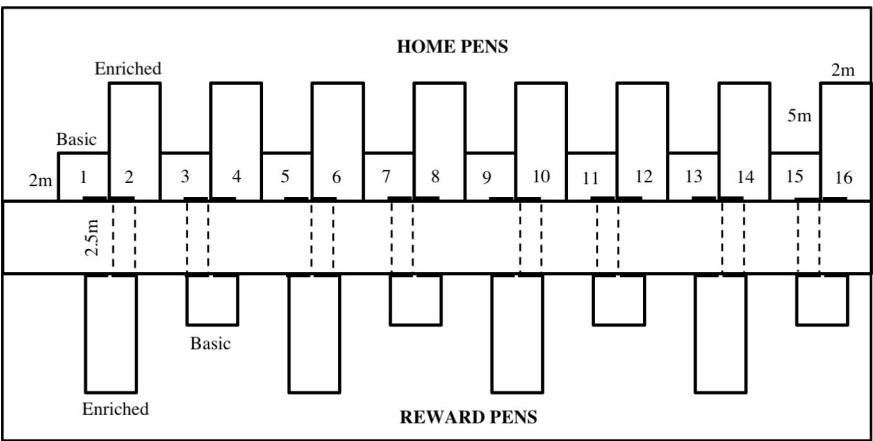

**Fig 2. Layout of experimental pens.** The barn contained home pens (basic or enriched) and reward pens (basic or enriched). Basic pens (2 m$^2$ per calf) were bedded with river stones. Enriched pens (5 m$^2$ per calf) were bedded with wood shavings and contained 3 'enrichment' items: automated grooming brush, hanging manila rope, and pile of straw. All home pens contained meal, hay and water feeders. Home pens were linked with an alleyway to the reward pen during conditioning, testing and contrast phases for 30 min per day.

to the opposite reward pen treatment; this created a condition where calves experienced an unexpected change in reward quality (e.g., reward loss or reward gain), such that calves previously receiving the basic reward pen now received the enriched reward pen and vice versa. The conditioning procedure in the contrast phase remained the same as described for the test phase.

## Behavioural observations

Fig 1A provides an overview of the behavioural observation periods in the test and contrast phases. Video cameras recorded the calves' behaviour in the home pen, exit door, alleyway and reward pen (DS-2CD2432F-I(W), HikVision, Zhejiang, China) using HikVision NVRs and software (DS-7732N1-14/16P and DS-771NI-SP, HikVision, Hangzhou, China). Each calf was observed continuously in the 3 min before the reward signal (*before* period), and in the 1 min during the reward signal (*anticipatory* period). Frequency and duration of behaviours, including behaviours directed at the light and door, play behaviours, and other behaviours, were scored following an ethogram (Table 1). After the exit door opened to provide access to the

**Table 1. Ethogram of anticipatory behaviours of dairy calves.**

| Behaviour | Description |
|---|---|
| **Behaviours directed at light and door** | |
| Looking at light | Head oriented toward flashing light while lying or standing |
| Looking at door | Head oriented toward door and in upright position while lying or standing |
| Touch light | Nose or head in contact with (or within 10 cm) of light |
| Touch door | Nose or head in contact with (or within 10 cm) of door |
| **Play behaviours** | |
| Locomotory play | Jumping: both forelegs off the ground and extended forwards (number of events); Running: calf trotting (2 beats) or galloping (3 beats) across or around the enclosure (number of events). If occurs within half body length of brush/rope, score as object play. If more than half body length, score as locomotive play. |
| Object play | Butting (head in contact with) brush/rope, or 'mock butt' where head is oriented downward and toward but not in contact with brush/rope, may be accompanied by locomotive play but if occurs within half body length of brush/rope then score as object play. Must be at least 2 s in between object play bouts to be scored as 2 separate bouts. |
| **Other behaviours** | |
| Lying inattentive | Lying down with head in upright position, without looking at light or door |
| Standing inattentive | Fully upright position, not moving body in any direction without looking at light or door. Defecation/urination may occur. |
| Walking | Lifting at least two legs simultaneously and moving the body in any direction, excluding locomotory play |
| Exploring | Nose is in contact with wall or floor of the pen. |
| Eating | Head in or above food/water trough |
| Brush use | Contact of any body part with brush |
| Rope use | Contact of any body part against rope |
| Groom | Scratching or grooming body with head |
| **Latency to access reward pen** | Duration from moment the home pen door opened to the moment the calf's rear legs crossed into the alleyway outside the home pen. |

Behaviours were recorded before, during and after signaling the arrival of a reward (access to either a basic or enriched reward pen for 30 min). The interval between the end of the signal (conditioned stimulus) and reward presentation (unconditioned stimulus) was 1 min.

reward pen (*access* period), latency to leave the home pen and behaviours expressed in the reward pen were recorded. One observer scored the *before* and *anticipatory* periods, and another observer scored the *access* period for descriptive purposes; inter- and intra-observer reliability were high ($\kappa > 0.90$). Due to technical issues with video recording, we experienced significant data loss resulting in only half of the original number of pairs having a complete dataset for the *before* and *anticipatory* periods (total 16 pairs; n = 8 with basic home pen; n = 8 with enriched home pen; n = 7 with basic reward pen; n = 9 with enriched reward pen. The number of pairs in each treatment combination were: B-B (n = 4), B-E (n = 4), E-B (n = 3) and E-E (n = 5)). Furthermore, for the *access* period, only three of each basic or enriched reward pen could be observed, so behavioural data from these groups was not modelled but is presented for descriptive purposes.

## Statistical analyses

All analyses were performed with SAS (version 9.3; SAS Inst. Inc., Cary, NC, USA) using calf within pair as the experimental unit. For each calf, total duration of each behaviour for each day was converted to a percentage of observation time during the *before* period (3 min) and *anticipatory* period (1 min). Behaviours in each period were summarized into the following behavioural categories (outlined in Table 1): total duration of behaviours directed at light and door, total duration of play behaviours, and total duration of other behaviours. We also calculated the total frequency of changes in behaviours (i.e. total number of behavioural transitions) occurring per min in the *before* and *anticipatory* periods. For example, the following sequence of behaviours was scored as 3 behavioural transitions: looks at the light while lying, then looks at the light while standing, then touches door, then shows locomotory play. There were no, or very few, cases of object play, brush use and rope use in these two periods, and therefore they were excluded from further analysis. Latency to leave the home pen at the start of the *access* period was calculated. Behaviours occurring in the *access* period were summarized as percentage of total duration in the reward pen but were not included in further analysis due to video loss.

All variables were screened using the UNIVARIATE procedure using box, distribution and probability plots. Total number of behavioural transitions was normally distributed, and variables that were transformed to achieve normality included: total duration of behaviours directed at light and door (square-root), total duration of other behaviours ($\log_{10}$), and latency to access reward pen in test phase ($\log_{10}$). The variable 'total duration of play behaviours' could not be transformed to achieve a normal distribution and thus was subjected to non-parametric analyses.

**Test phase analyses.** To determine if calves showed anticipatory behaviours (i.e., total behavioural transitions, total duration of behaviours directed at light and door, and total duration of other behaviours) in the *anticipatory* period compared to the *before* period, we used a generalized linear mixed model with repeated measures (PROC MIXED) with the following fixed effects: period (before or anticipatory), home pen (basic or enriched), reward pen (basic or enriched), rep (1 or 2), the interaction of home pen and reward pen, and the three-way interaction of home pen, reward pen and period. Test day within period was specified as a repeated measure, and calf within pair was specified as the subject and an autoregressive covariance structure was the best fit according to lowest AIC. We conducted a Friedman test to analyze for differences in play behaviour between the two periods, controlling for home pens (basic or enriched), reward pens (basic or enriched), and test day (day 13, 14, or 15).

To determine if anticipatory behaviours and latency to access the reward pen differed between home and reward pen treatments, and whether these changed throughout the test phase, we conducted a generalized linear mixed model with repeated measures (PROC

MIXED) with the following fixed effects: home pen (basic or enriched), reward pen (basic or enriched), test day (continuous variable), rep (1 or 2), and the interaction of home pen and reward pen. Test day was specified as a repeated measure with calf within pair as the subject and an autoregressive covariance structure was the best fit according to lowest AIC. We also conducted a Friedman test to analyze for differences in play behaviour between home pens (basic or enriched) and reward pens (basic or enriched), controlling for test day (day 13, 14, 15).

**Contrast phase analyses.** To determine if the home or reward pen treatment affected how calves responded to a reward loss or reward gain, anticipatory behaviours and latency to access the reward were compared within and between treatments before the reward pen change (average of day 13 to 15 in test phase, when calves were expecting the original reward pen) and after the change, when calves expected the new reward pen (day 17, first day of contrast phase; average of day 18 to 20, subsequent days of contrast phase). Day 16 was considered a transition day when the reward pen changed and resulted in disruptions to the calf's routine, and thus was excluded from analysis. The model (PROC MIXED) included the fixed effects of home pen (basic or enriched), original reward pen (basic or enriched), stage of change (3 levels: before reward change, first day after change, subsequent days after change), rep (1 or 2), and the three-way interaction of home pen, original reward pen and stage of change. Stage of change was specified as a repeated measure with calf within pair as the subject and an autoregressive covariance structure was the best fit according to AIC. The Friedman test was used to analyze for differences in play behaviour between home and reward pen treatments for each stage of change.

**Results reporting.** For all test and contrast period models, transformed variables (total duration of behaviours directed at light and door and total duration of other behaviours) were also modeled in their original scale and no differences in model outcomes were found; thus, all results are reported as least squares means ± SE for frequency of behavioural transitions, duration of light and door behaviours, other behaviours, and latency to access reward pen in their original scale. The Cochran-Mantel-Haenszel statistic and raw means ± SE are reported for duration of play behaviours from the non-parametric analyses. Significance was declared at $P \leq 0.05$. For the access period, where limited sample size prevented modeling, the behaviours that calves expressed after accessing the reward pen are presented as percentage of total duration (± SD) in the reward pen for the test phase (average of day 13 to 15), for the first day of the contrast phase (day 17, the day after a change in reward pen quality), and the subsequent days of the contrast phase (average of day 18 to 20).

## Results

### Test phase

Calves changed their behaviour when access to the reward pen was signaled. During the *anticipatory* period (compared to *before*) and across all treatments, there was a 10-fold increase in the number of behavioural transitions (Fig 3A; $F_{1,27} = 263.5$, $P < 0.001$), a 6-fold increase in the duration of behaviours directed at the light and door signal (Fig 3B; $F_{1,27} = 347.3$, $P < 0.001$), and a 50% decrease in the duration of other behaviours (Fig 3C; $F_{1,27} = 338.0$, $P < 0.001$). No play behaviours were observed in the *before* period, but play increased during the *anticipatory* period for all treatments except B-E (raw means ± SE: 0.17 ± 0.26, 0.52 ± 0.68 and 0.38 ± 0.42% duration of period, for B-B, E-B, and E-E treatments, respectively; Cochran-Mantel-Haenszel row mean scores difference = 5.1; $P = 0.02$).

Behaviours expressed during the *anticipatory* period for home pen and reward pen treatments are presented in Table 2. During the *anticipatory* period, calves in the basic home pen treatment showed more behavioural transitions compared to calves in the enriched home pen

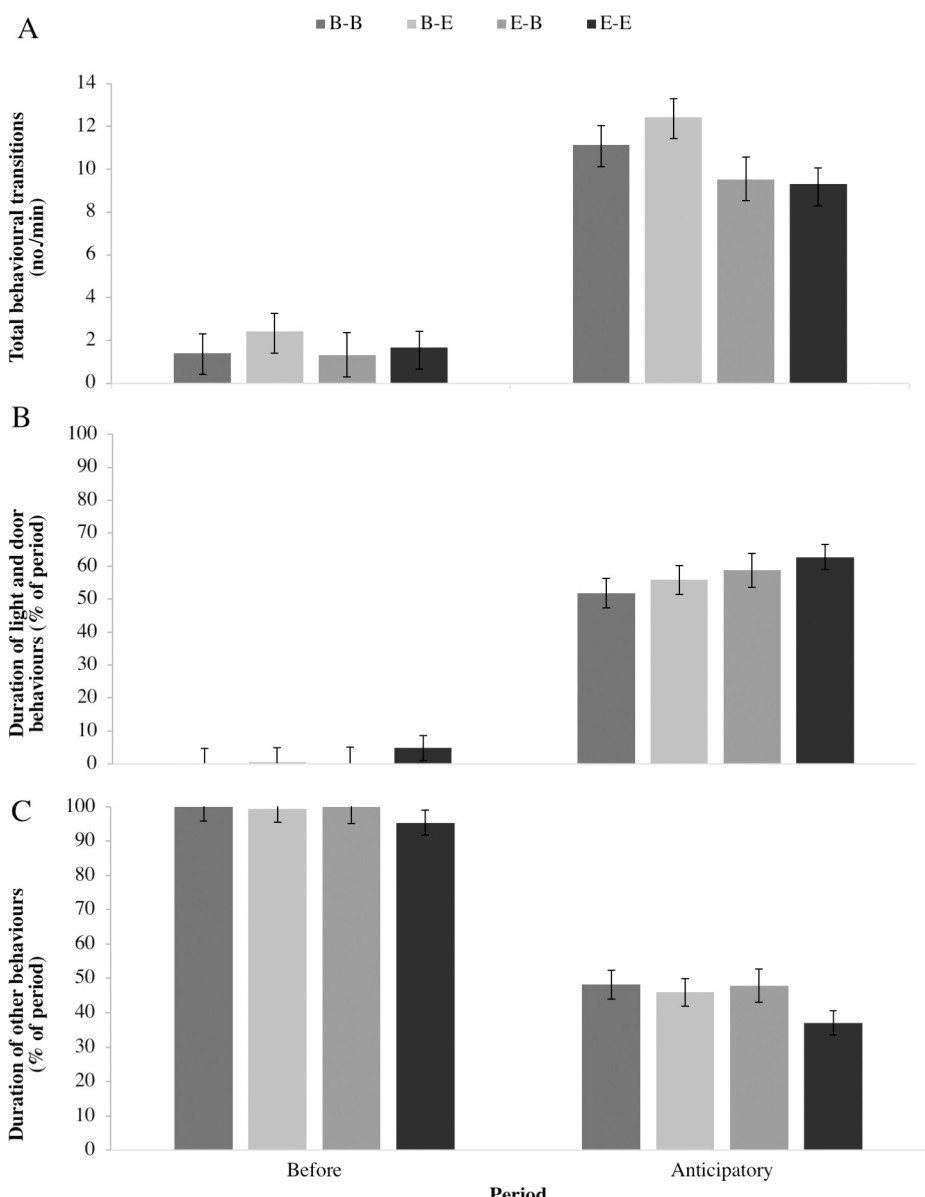

**Fig 3. Behaviours expressed before and during the anticipatory periods.** Behaviours expressed before (3 min before presentation of the conditioned stimulus, light turned on) and during the anticipatory periods (1 min interval between presentation of the conditioned stimulus, and the unconditioned stimulus, access to the reward pen) for each treatment (2 x 2 factorial: basic or enriched home pen, with access to basic or enriched reward pen). Least squares means ± SE for (A) frequency of behavioural transitions (no./min), (B) duration of light and door behaviours (% of period) and (C) duration of other behaviours (% of period).

treatment ($F_{1,27}$ = 5.2; $P$ = 0.03), but there was no difference in reward pen treatment ($F_{1,27}$ = 0.31; $P$ = 0.58) or the interaction of home and reward pen treatments ($F_{1,27}$ = 0.18; $P$ = 0.68). There were no differences between home pen or reward pen treatments, or their interaction for duration of behaviours directed at light and door ($F_{1,27}$ < 1.2, $P$ > 0.28), play behaviours (Cochran-Mantel-Haenszel row mean scores difference < 2.3; $P$ > 0.13), or other behaviours ($F_{1,27}$ < 1.6; $P$ > 0.22). Duration or frequency of behaviours in the *anticipatory* period did not differ across test days ($F_{1,50}$ < 0.40; $P$ > 0.53).

**Table 2. Behaviours expressed during the anticipatory period for each treatment.**

| Variable | Treatment | | | | | | Interaction |
| | Home pen | | | Reward pen | | | |
| | Basic | Enriched | P-value | Basic | Enriched | P-value | P- value |
|---|---|---|---|---|---|---|---|
| Total behavioural transitions (no./min) | 11.5 ± 0.77 | 9.2 ± 0.81 | 0.03 | 10.0 ± 0.90 | 10.6 ± 0.70 | 0.58 | 0.68 |
| Duration of light and door behaviours (% of period) | 52.0 ± 4.6 | 58.6 ± 4.8 | 0.28 | 52.6 ± 5.3 | 58.0 ± 4.1 | 0.39 | 0.89 |
| Duration of other behaviours (% of period) | 48.9 ± 4.6 | 44.4 ± 4.8 | 0.45 | 50.5 ± 5.3 | 42.7 ± 4.1 | 0.22 | 0.35 |
| Duration of play behaviours (% of period) | 0.08 ± 0.13 | 0.44 ± 0.36 | 0.32 | 0.32 ± 0.31 | 0.21 ± 0.25 | 0.40 | N/A [1] |

Behaviours expressed during the 1 min interval between presentation of the conditioned stimulus, and the unconditioned stimulus of access to the reward pen) for home pen and reward pen treatments, and their interaction (2 x 2 factorial: basic or enriched home pen, with access to basic or enriched reward pen). Results are presented as least squares means ± SE for frequency of behavioural transitions (no./min), duration of light and door behaviours (% of period) and duration of other behaviours (% of period). Raw means ± SE are presented for duration of play behaviours (% of period) due to non-parametric analysis.

[1] Interaction term for home and reward pen not available using non-parametric analysis with Friedman test.

Latency to access the reward pen was shorter for calves in the basic versus enriched home pen treatment (B: 5.0 ± 2.7 s; E: 21.3 ± 2.8 s; $F_{1,27}$ = 21.3; $P < 0.001$), but was not affected by reward pen treatment (B: 11.4 ± 3.1 s; E: 15.0 ± 2.4 s; $F_{1,27}$ = 0.98; $P = 0.33$) or the interaction of home and reward pen treatments ($F_{1,27}$ = 0.44; $P = 0.51$). Latency to access the reward pen did not differ across test days ($F_{1,50}$ = 0.35; $P = 0.56$).

## Contrast phase

In the contrast phase, the quality of the reward pen changed from basic to enriched, or vice versa, such that on day 13 to 15 calves expected the original reward pen treatment (before change) and on day 17 to 20 calves expected the new reward pen treatment (after change) (see Fig 1B). There was a 3-way interaction of home pen, original reward pen and stage of change for total number of behavioural transitions ($F_{7,51}$ = 3.5; $P < 0.01$). In line with our hypothesis, basic-housed calves experiencing reward loss (B-E calves changed to a basic reward pen) decreased the frequency of behavioural transitions from before to after the reward pen change (Fig 4A; $t_{1,52}$ = 2.5, $P = 0.01$). The opposite was observed for enriched-housed calves experiencing reward loss (E-E calves changed to basic reward pen) who showed an increase in this behaviour ($t_{1,52}$ = 2.6; $P = 0.01$). There was no change in frequency of behavioural transitions in response to reward gain for basic-housed calves (B-B calves changed to an enriched reward pen; $t_{1,52}$ = 1.7, $P = 0.09$) or for enriched-housed calves (E-B calves changed to enriched reward pen; $t_{1,52}$ = 1.4; $P = 0.16$).

Other anticipatory behaviours, the duration of light and door behaviours (Fig 4B) and duration of other behaviours (Fig 4C), did not change after the reward pen change for either home pen or original reward pen treatments (light and door behaviours: $F_{2,51} < 0.75$, $P > 0.48$; other behaviours: $F_{2,51} < 1.4$, $P > 0.26$).

Play behaviours differed between treatments before and after the reward pen change (Cochran-Mantel-Haenszel row mean scores difference = 9.1, $P = 0.03$); this difference was driven by the increase in play behaviour for enriched-housed calves experiencing reward loss (E-E calves changed to basic reward pen) from before to after reward pen change (raw means: 0.41 ± 0.28 versus 1.3 ± 0.52% duration of period, respectively), while all other treatments showed no play behaviour on the day after the change (raw means: B-B: 0.21± 0.21 versus 0 ± 0; B-E: 0 ± 0 versus 0 ± 0; E-B: 0.65 ± 0.40 versus 0 ± 0% duration of period).

The change in latency to access the reward pen differed depending on home pen treatment (Fig 5; $F_{2,51}$ = 9.9; $P < 0.001$); basic-housed calves showed no change in latency ($t_{1,51}$ = 0.05;

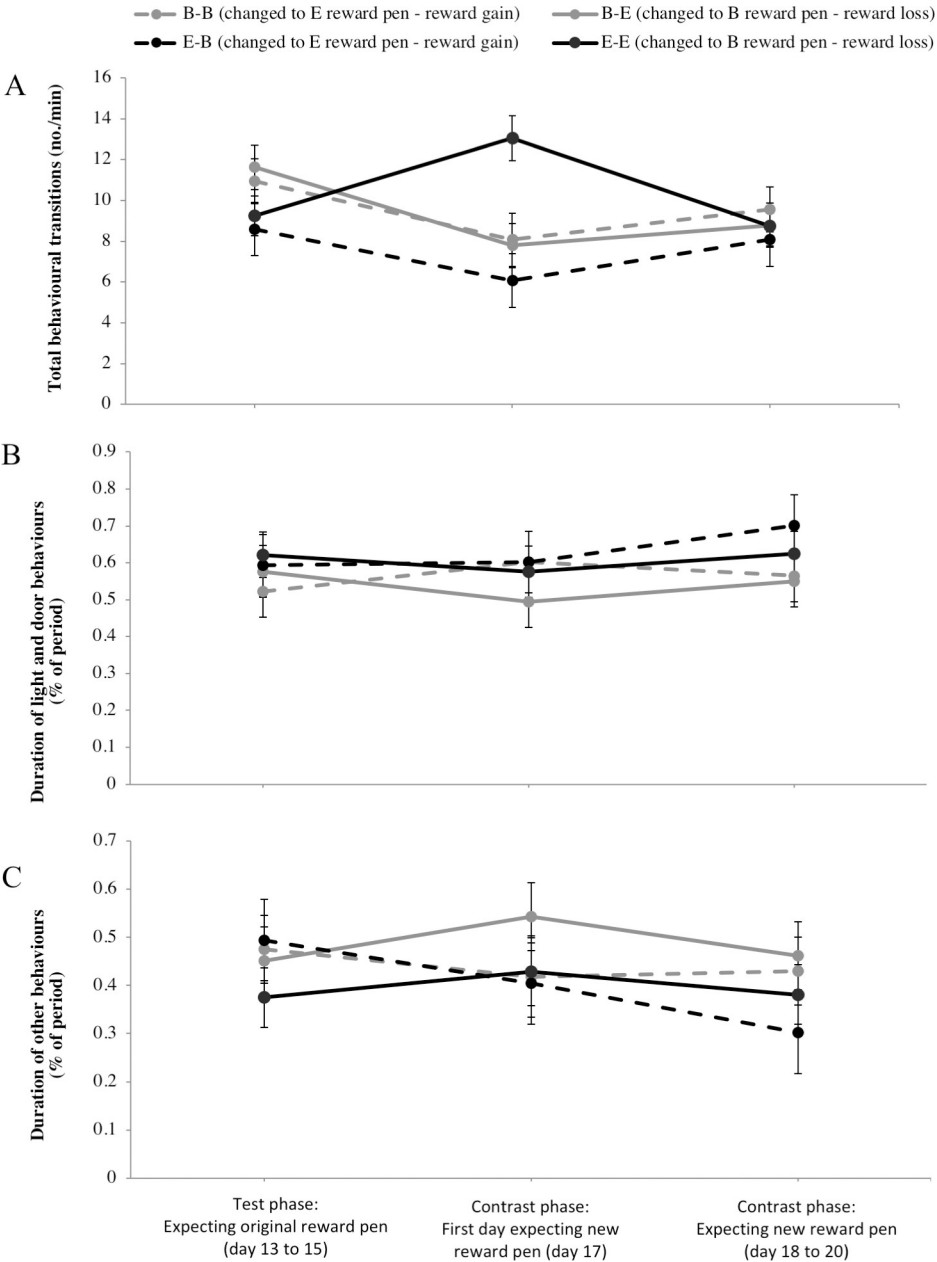

**Fig 4. Anticipatory behaviour before and after an unexpected change in reward pen quality.** Before the change (day 13 to 15 of test phase), calves entered the reward pen expecting the original reward pen. After the change (on day 17, the first day after the change, and on subsequent days 18 to 20), calves entered the reward pen expecting the new reward pen. Calves that changed from an enriched to basic reward pen were expected to experience a reward loss, while calves that changed from a basic to enriched reward pen were expected to experience a reward gain. Least squares means ± SE of (A) frequency of behavioural transitions (no./min), (B) duration of light and door behaviours (% of period), and (C) duration of other behaviours (% of period).

$P = 0.95$) while enriched-housed calves decreased latency ($t_{1,51} = 5.0$; $P < 0.001$) after the reward pen change. However, latency to access the new reward pen did not depend on whether the original reward pen was basic or enriched ($F_{2,51} = 1.2$; $P = 0.32$), indicating that latency was not affected by the experience of reward loss or gain.

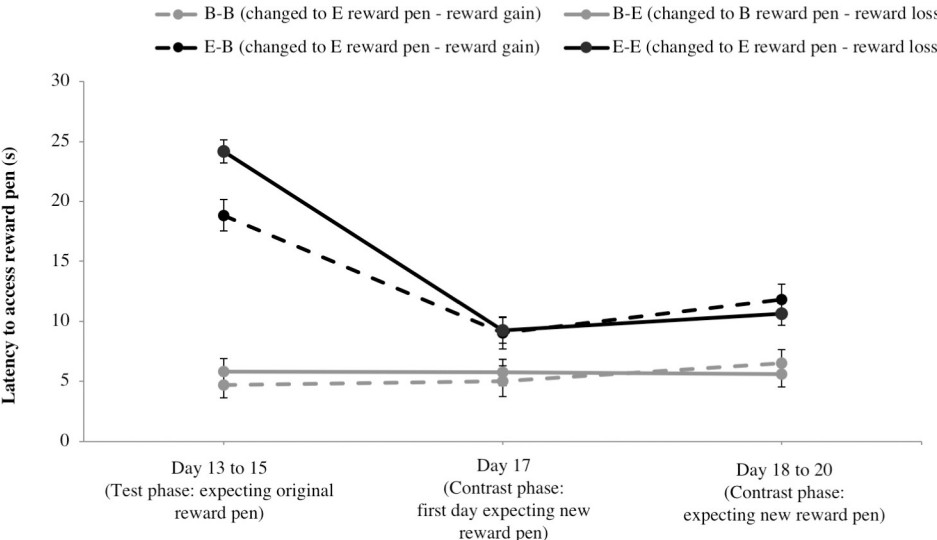

**Fig 5. Latency to access the reward pen before and after an unexpected change in reward pen quality.** Before the change (day 13 to 15 of test phase), calves entered the reward pen expecting the original reward pen. After the change (on day 17, the first day after the change, and on subsequent days 18 to 20), calves entered the reward pen expecting the new reward pen. Calves that changed from an enriched to basic reward pen were expected to experience a reward loss, while calves that changed from a basic to enriched reward pen were expected to experience a reward gain. Results shown as least squares means ± SE.

## Access period

The behaviours expressed after accessing the reward pen in the test and contrast phases (in the 30 min *access* period) are presented for descriptive purposes in S1 Table.

## Discussion

The primary aim of this study was to investigate how anticipatory behaviour of dairy calves is affected by housing environment and reward quality. Calves housed in a basic environment expressed more anticipatory behaviour for a reward compared to calves housed in an enriched environment. Greater anticipatory behaviour, measured as the number of behavioural transitions, has been observed in rats housed in standard environments compared to enriched and semi-natural environments [16, 17], and suggests an increased sensitivity to rewards. The latency to reach or consume a reward is also indicative of reward sensitivity [14]; our basic-housed calves were quicker to access the reward pen compared to enriched-housed calves. Similarly, hens that experienced short-term isolation ran faster for mealworms than control hens [30]; such increased sensitivity to rewards has been proposed to reflect a negative emotional state [6]. Indeed, bottlenose dolphins that showed more anticipatory behaviour also showed a pessimistic bias in a judgement bias task, a common method of assessing negative mood states (i.e. a longer term emotional state) in animals [31].

However, severely depressed individuals experiencing chronic stress may respond in the opposite manner, demonstrating decreased sensitivity to reward. For instance, rats experiencing repeated social defeat or long-term social isolation reduced, or failed to have any, anticipatory response to a sucrose reward [18, 32], and pigs raised in barren housing ran slower for pieces of apple than pigs raised in enriched housing [23]. These studies suggest that anticipatory behaviours and latency to reach a reward are influenced by the animal's previous experience and underlying emotional state. Therefore, our findings suggest that basic-housed calves may have

experienced a negative emotional state, but not one of severe depression. All calves in our study were housed in pairs, which may have reduced the negative impact of the basic housing conditions compared to if calves had been housed in isolation (a common practice in the global dairy industry). Given that long-term social isolation negatively affected anticipation for reward in other species, it would be valuable to investigate how individual housing impacts the emotional state of calves in order to better understand the impact of this practice on calf welfare.

To our knowledge, only two other farm animal studies have investigated anticipatory behaviour in different housing systems (chickens [13] and mink [11]); neither study found differences in anticipation for a food reward between basic and enriched housing environments. One explanation for the difference between their studies and ours is that we had a substantial contrast between housing environments. Our enriched pen provided 2.5 times more space, a comfortable bedding surface and items to promote grooming and play, compared to the basic pen, which represented minimal industry housing practices for dairy calves in New Zealand [28]. It is unknown which specific aspects of our housing conditions contributed to differences in anticipatory behaviour (and thus inferred differences in emotional state); it was not the purpose of the study to tease out the relationships between these aspects, but rather they were deliberately chosen in combination to create very different environments. Furthermore, we tested anticipation of calves to access either a basic or enriched reward pen, while a food reward was used in the other studies. The type of reward elicits differences in expression and type of anticipatory behaviours; for instance, chickens expressed more pecking and pushing at a door for access to a dust-bathing substrate compared to a food reward [21], and lambs explored and walked more before receiving a food reward compared to an opportunity to play [9]. These studies, including our own, provide an opportunity to understand how individuals perceive the incentive value of different rewards.

We expected that calves would express more anticipatory behaviour for access to the enriched versus the basic reward pen (indicating that the enriched pen has more perceived value), but that basic-housed calves would show more anticipation for the enriched reward pen than enriched-housed calves (indicating that basic-housed calves are more sensitive to highly valued rewards). However, our calves showed a similar level of anticipation for the basic and enriched reward, although basic-housed calves showed more anticipatory behaviour than enriched-housed calves overall. Latency to access the reward also did not differ between reward pens. In contrast, Van Der Harst et al. [20] found that standard-housed rats showed more anticipation for transfer to an enriched cage versus transfer to another standard cage, but transfer to another standard cage induced greater expression of some behavioural elements than staying in the same home pen. The authors suggested that transfer to a standard cage had intermediate rewarding value due to the anticipation of simply being transferred out of the home cage. This finding may also explain why we did not find differences in anticipation for enriched and basic reward pens; the opportunity for calves to leave their home pen may have been rewarding in itself. Our descriptive analysis of behaviours that were expressed in the reward pens suggested that both basic and enriched pens provided an increase in space and opportunity for calves to play and explore; for instance, the basic reward pen offered opportunity to explore new river stone bedding while the enriched reward pen offered opportunity to explore new grooming materials and wood chip bedding. Thus, the novelty of the pens in different respects may have led to similarities in anticipation for access to these pens. The fact that basic-housed calves showed more anticipation than enriched-housed calves for either reward pen suggests that these opportunities may be more valuable to these calves, possibly due to behavioural deprivation in their basic home pen. This is supported by a recent study showing that rats housed in standard cages exhibited greater initial exploration of a new environment, suggesting that individuals in deprived environments have a higher propensity to

explore novelty [33]. A limitation of the basic environment is that calves were unable to perform the same variety of anticipatory behaviours as the enriched-housed calves, so the energy requirements to change between behaviours, or the value of performing the different behaviours, may be different between basic- and enriched-housed calves. Finally, we acknowledge that the significant loss of data in our study resulted in low sample size, which may have limited the power to detect a difference in anticipatory behaviour and latency to access reward for the interaction between home and reward pen environments. It also may have limited our ability to identify differences in play behaviours, especially in basic-housed calves; therefore, we encourage others to replicate this work to investigate why calves appeared to value basic and enriched reward pens similarly.

An additional aim of this study was to investigate if anticipatory behaviour and latency to access the reward changed in response to an unexpected reward loss or gain, and if these behaviours differed depending on the calves' housing environment. As hypothesized, basic-housed calves decreased their frequency of behavioural transitions when experiencing reward loss (when changed from an enriched to a basic reward pen). A negative response to a discrepancy in expectation is suggested to indicate an underlying negative mood state [24, 34]; negative responses are strongest in rats housed in barren environments [14], and in pigs experiencing a loss of enriched housing [23]. Thus, our results suggest that basic-housed calves may experience a negative emotional state given their negative response to reward loss. In contrast, an unexpected finding was that enriched-housed calves increased the frequency of behavioural transitions in response to reward loss (when changed from an enriched to a basic reward pen). Enriched-housed calves also accessed the reward pen faster after the change, regardless of its quality, compared to basic-housed calves. Hypersensitivity to increases in reward value have been shown in animals experiencing longer-term positive emotional states (i.e., positive mood state) due to heightened activity of the dopaminergic system [7, 35]. However, it remains to be understood how strong these positive or negative mood states are experienced by calves in our basic and enriched pens; we acknowledge that factors other than housing may be influencing long term mood state (e.g., separation from the dam [36]).

We found that calves experiencing a reward gain (or 'upshift') showed no change in anticipatory behaviour or latency to access the reward, regardless of housing environment. This finding is consistent with the literature showing limited behavioural changes during a reward upshift relative to unshifted controls; the lack of effects are often attributed to either the novelty of the upshifted reward or to ceiling effects [37, 38]. The latter is a possibility in our study given that basic-housed calves accessed the basic reward pen very quickly (within 5 s) before the change and may have been physically unable to access the reward pen faster after the change. The former novelty effect may arise from neophobia (e.g., fear of novel situations or foods) when the reward quality increases, leading to initial suppression of reward-directed behaviours that return to levels similar to unshifted controls [37]. For instance, lambs decreased the frequency of an operant task when a food reward size increased [39], and decreased the frequency of behavioural transitions when a food reward changed to an opportunity to play [9]. A drawback of our study design in the contrast phase was the lack of an unshifted control group, so our conclusions regarding how basic- and enriched-housed calves perceive reward loss or gain remain cautious. It may be that our mixed findings are attributed to individual differences in how calves experienced the change in reward pen quality; we had considered the enriched pen to be of higher absolute quality, but other factors such as novelty and opportunities to engage in different behaviours in different pen types may have contributed to how individuals valued these pens.

An increase in activity or behavioural transitions is one of the most common observations across anticipatory studies of animals and appears to be expressed more when the reward is

perceived to be positive. For example, rats showed a large increase in frequency of behavioural transitions before transfer to an enriched cage or sexual contact compared to transfer to a standard cage or a forced swimming session [20], and goats were more active when anticipating accessible food versus inaccessible food [40]. However, increased activity may also indicate anticipation of a negative event, as reported in chickens [41], or may reflect frustration, especially when the interval between the conditioned stimulus (signal) and unconditioned stimulus (reward delivery) is prolonged [42] (see review by Anderson et al. [19]). In our study, this interval was relatively short (1 min), and included a training period that gradually developed the association between the conditioned and unconditioned stimulus over 12 days; these features are expected to minimize the development of frustration-related behaviours during the anticipatory period. Therefore, we believe that the anticipatory behaviour expressed by our calves likely reflected positive anticipation for a reward that differed depending on the home environment. Our findings also more broadly support the growing literature indicating that dairy calves are sentient animals that make associations with and hold expectations for future events.

## Conclusions

Dairy calves in basic housing showed more anticipatory behaviour for access to a reward pen than calves in enriched housing, but this behaviour did not depend on whether the reward pen was basic or enriched. Basic-housed calves also showed a decrease in anticipatory behaviour when the reward quality was unexpectedly reduced. Our findings suggest that calves housed in restricted conditions may be more sensitive to presentation of rewards and changes in their quality; thus, these calves may have been experiencing a more negative emotional state compared to calves in an enriched environment. This finding emphasizes that the emotional state of calves is affected by their housing conditions and reinforces the importance of housing quality for calf welfare. However, we caution that significant data loss in our study limited our capacity to report behaviour during the reward period, and thus limits our ability to extrapolate on long-term mood state. We encourage future work to understand the aspects of housing conditions that contribute to differences in emotional state.

## Supporting information

**S1 Table. Behaviours expressed in the reward pen.** Following a 1 min anticipatory period in the home pen, behaviours expressed during the 30 min access period in the reward pen (basic or enriched, n = 3 each). Behaviours are presented as percentage of total duration (± SD) in the reward pen for the test phase (average of day 13 to 15), for the day after a change in reward pen quality[1] (day 17), and for subsequent days of the contrast phase (average of day 18 to 20). Data is presented for descriptive purposes only due to data loss.
(DOCX)

## Acknowledgments

Facility setup was made possible by the skilled work of Stu Eaton and his team. We are grateful for AgResearch and Tokanui Dairy Research Farm personnel for their help during the study, especially Ian Johnston, Trevor Watson, Mhairi Sutherland, and Luca van Dijk. Thank you also to Laura Counsell and Lauren Little for video watching.

## Author Contributions

**Conceptualization:** Heather W. Neave, Gosia Zobel.

**Data curation:** Heather W. Neave.

**Formal analysis:** Heather W. Neave, Gosia Zobel.

**Funding acquisition:** James R. Webster.

**Investigation:** Heather W. Neave, Gosia Zobel.

**Methodology:** Heather W. Neave, Gosia Zobel.

**Project administration:** James R. Webster.

**Resources:** James R. Webster.

**Supervision:** James R. Webster, Gosia Zobel.

**Visualization:** Gosia Zobel.

**Writing – original draft:** Heather W. Neave.

**Writing – review & editing:** Heather W. Neave, James R. Webster, Gosia Zobel.

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
