## [Decision Letter · Decision Letter 0]

26 Nov 2020

PONE-D-20-32202

Anticipatory behaviour as an indicator of the welfare of dairy calves in different housing environments

PLOS ONE

Dear Dr. Zobel,

Thank you for submitting your manuscript to PLOS ONE. After careful consideration, we feel that it has merit but does not fully meet PLOS ONE’s publication criteria as it currently stands. Therefore, we invite you to submit a revised version of the manuscript that addresses the points raised during the review process.

The paper could have an important technical contribution in the area. I agree with all reviewers think that mainly the manuscript have several technical deficiencies. There are a lot of careless mistakes in the manuscript and the whole manuscript shall be reviewed by the authors carefully.

We look forward to receiving your revised manuscript.

Kind regards,

Arda Yildirim, Ph.D.

Academic Editor

PLOS ONE

Journal Requirements:

3.Thank you for stating the following in the Financial Disclosure section:

[The study funding was provided by AgResearch Core funding (A19041). HN was supported by the Natural Science and Engineering Research Council Canada Graduate Scholarship and the Michael Smith Foreign Study Supplement.]. 

We note that one or more of the authors have an affiliation to the commercial funders of this research study : AgResearch Ltd.

Additional Editor Comments (if provided):

Please check the manuscript thoroughly for the English language and correct, as required. For your guidance, you can check the reviewers' comments. Thank you for giving us the opportunity to consider your work.

Reviewers' comments:

Reviewer's Responses to Questions

**Comments to the Author**

1. Is the manuscript technically sound, and do the data support the conclusions?

Reviewer #1: Yes

Reviewer #2: Yes

2. Has the statistical analysis been performed appropriately and rigorously? 

Reviewer #1: Yes

Reviewer #2: I Don't Know

3. Have the authors made all data underlying the findings in their manuscript fully available?

Reviewer #1: Yes

Reviewer #2: Yes

4. Is the manuscript presented in an intelligible fashion and written in standard English?

Reviewer #1: Yes

Reviewer #2: Yes

5. Review Comments to the Author

Reviewer #1: This is a nicely written paper for the most part, and provides a clear and thorough explanation of the work done. The study is very interesting and tackles a relevant and important area of dairy calf welfare, through using anticipatory behaviour. It is a real shame that, after all the care and attention given to the study design and execution, that the very significant loss of video data has reduced the value and impact of the work. Nevertheless there is enough novel work in here to merit publication, although the outcomes and understanding from the study remain rather weaker than would otherwise have been the case. Overall I have only minor comments regarding some rather wayward use of punctuation in places, and where the clarity of the study could be improved:

Line 13: End sentence at animals and begin new sentence with However

Line 16: Assume this is the data from the 16 pairs of calves for which you did have data rather than the 64 animals you began the study with?

Line 23: you variously describe this metric as shifts, changes or transitions in behaviour. Since you also use shifts for 'upshifts' in reward value it would be better to firstly be consistent and secondly probably to stick to either transitions or changes throughout (transitions probably best)

L24: no effect 'on anticipation'

L27: end sentence at behaviour, being new sentence with However

L41: end sentence at [4,5], begin new sentence with This

L45: end sentence at reward, begin new sentence with This

L48: end sentence at reward and provide reference; begin new sentence with Thus

L52-55: referencing seems to have changed to a different style? Also in a few other places in the manuscript

L89: end sentence at reward, begin new sentence with However, this...

L142: suggest change 'had to' to 'could' since you state at L148 that calves were not forced

L167: E-B

L212-217: This is such a shame! At L214-215 can you clarify the animal numbers in terms of B-B, B-E etc?

L238: not clear how latency to leave was defined - add to ethogram?

L239-240: clarify this was related to data loss and lack of data not any other decision - these data would have been really beneficial to the study and understanding the responses sadly.

L449: reference style?

L469-474: Yes, would have really added value to this paper to know what the calves did in the pens to reduce some of the speculation

L487-489: Although the enriched pens were better than the barren pens I would be careful in inferring that the animals were in an unrelieved state of positive welfare, it may be that it was just a bit better. After all they have been separated from their dam, which has been shown to influence judgement bias, and the enriched pens is far from optimal for young calves.

L501: reference style?

Reviewer #2: Manuscript should be checked carefully for english and grammatical error. Hypothesis is not clear. Introductory section require revision and only relevant information needs to be incorporated. Discussion section needs more rectification.

6. PLOS authors have the option to publish the peer review history of their article (what does this mean?). If published, this will include your full peer review and any attached files.

Reviewer #1: No

Reviewer #2: No

---

## [Author Response · Author response to Decision Letter 0]

2 Dec 2020

Response to Reviewers – PloS ONE. Anticipatory behaviour as an indicator of the welfare of dairy calves in different housing environments

Journal Requirements:

AU: Corrected

AU: Corrected

3.Thank you for stating the following in the Financial Disclosure section:

[The study funding was provided by AgResearch Core funding (A19041). HN was supported by the Natural Science and Engineering Research Council Canada Graduate Scholarship and the Michael Smith Foreign Study Supplement.]. 

We note that one or more of the authors have an affiliation to the commercial funders of this research study : AgResearch Ltd.

AU: Corrected and added to coverletter as requested. 

AU: Corrected and added to coverletter as requested. 

Additional Editor Comments (if provided):

Please check the manuscript thoroughly for the English language and correct, as required. For your guidance, you can check the reviewers' comments. Thank you for giving us the opportunity to consider your work.

AU: Thank you for your review of our work. We have reviewed the manuscript and checked it carefully for English and grammar. We have also updated the formatting according to the formatting requirements. We apologize for this oversight.

Reviewers' comments:

Reviewer's Responses to Questions

Comments to the Author

1. Is the manuscript technically sound, and do the data support the conclusions?

Reviewer #1: Yes

Reviewer #2: Yes

2. Has the statistical analysis been performed appropriately and rigorously?

Reviewer #1: Yes

Reviewer #2: I Don't Know

3. Have the authors made all data underlying the findings in their manuscript fully available?

Reviewer #1: Yes

Reviewer #2: Yes

4. Is the manuscript presented in an intelligible fashion and written in standard English?

Reviewer #1: Yes

Reviewer #2: Yes

5. Review Comments to the Author

Reviewer #1: This is a nicely written paper for the most part, and provides a clear and thorough explanation of the work done. The study is very interesting and tackles a relevant and important area of dairy calf welfare, through using anticipatory behaviour. It is a real shame that, after all the care and attention given to the study design and execution, that the very significant loss of video data has reduced the value and impact of the work. Nevertheless there is enough novel work in here to merit publication, although the outcomes and understanding from the study remain rather weaker than would otherwise have been the case. Overall I have only minor comments regarding some rather wayward use of punctuation in places, and where the clarity of the study could be improved:

AU: We are very appreciative of the Reviewer’s positive feedback on the study. Indeed, we were absolutely gutted by our video system’s massive failure. We have since updated to a new server to avoid this kind of thing happening in the future. Nonetheless, we are grateful that Reviewers have seen merit in the data we were able to salvage. We’ve hopefully thoroughly addressed your comments but welcome any follow-up you may have!

Line 13: End sentence at animals and begin new sentence with However

AU: Changed.

Line 16: Assume this is the data from the 16 pairs of calves for which you did have data rather than the 64 animals you began the study with?

AU: Correct – we chose to just indicate the number of calves for which we had data in the abstract, in the interest of space. We indicate the original number of calves enrolled in the methods section.

Line 23: you variously describe this metric as shifts, changes or transitions in behaviour. Since you also use shifts for 'upshifts' in reward value it would be better to firstly be consistent and secondly probably to stick to either transitions or changes throughout (transitions probably best)

AU: Thank you for this suggestion. We have changed to transitions throughout.

L24: no effect 'on anticipation'

AU: Changed.

L27: end sentence at behaviour, being new sentence with However

AU: Changed.

L41: end sentence at [4,5], begin new sentence with This

AU: Changed. 

L45: end sentence at reward, begin new sentence with This

AU: Changed.

L48: end sentence at reward and provide reference; begin new sentence with Thus

AU: Changed and added reference.

L52-55: referencing seems to have changed to a different style? Also in a few other places in the manuscript

AU: Corrected.

L89: end sentence at reward, begin new sentence with However, this...

AU: Changed.

L142: suggest change 'had to' to 'could' since you state at L148 that calves were not forced

AU: Changed.

L167: E-B

AU: Corrected.

L212-217: This is such a shame! At L214-215 can you clarify the animal numbers in terms of B-B, B-E etc?

AU: Indeed – given the time and effort that went into this study, we were grateful to be able to salvage what we did! We have now added animal numbers for each treatment as well. 

L238: not clear how latency to leave was defined - add to ethogram?

AU: Thank you. This has been added to the ethogram.

L239-240: clarify this was related to data loss and lack of data not any other decision - these data would have been really beneficial to the study and understanding the responses sadly.

AU: We have added this clarification.

L449: reference style?

AU: Apologies, we have corrected the reference style throughout. 

L469-474: Yes, would have really added value to this paper to know what the calves did in the pens to reduce some of the speculation

AU: We agree this would have been ideal but unfortunately was not possible with our data loss.

L487-489: Although the enriched pens were better than the barren pens I would be careful in inferring that the animals were in an unrelieved state of positive welfare, it may be that it was just a bit better. After all they have been separated from their dam, which has been shown to influence judgement bias, and the enriched pens is far from optimal for young calves.

AU: This is a valid point – we have added some consideration around this point to the end of the sentence. 

L501: reference style?

AU: corrected

Reviewer #2: Manuscript should be checked carefully for english and grammatical error. Hypothesis is not clear. Introductory section require revision and only relevant information needs to be incorporated. Discussion section needs more rectification.

AU: We have carefully reviewed the manuscript for English and grammar, and have added a clarification regarding our hypothesis. We have cut back some of the introduction to be more focused. With regards to the discussion, while we acknowledge there is a lot of content, we feel it is all relevant to results found, and the story being told; we have therefore left most of it as it was, with the exception of a few changes requested by Reviewer 1. If however Reviewer 2 has some specific suggestions, we are happy to examine and address these.

---

## [Decision Letter · Decision Letter 1]

7 Jan 2021

Anticipatory behaviour as an indicator of the welfare of dairy calves in different housing environments

PONE-D-20-32202R1

Dear Dr. Zobel,

We’re pleased to inform you that your manuscript has been judged scientifically suitable for publication and will be formally accepted for publication once it meets all outstanding technical requirements.

Kind regards,

Arda Yildirim, Ph.D.

Academic Editor

PLOS ONE

https://www.researchgate.net/profile/Arda_Yildirim2

Additional Editor Comments (optional):

Reviewers' comments:

Reviewer's Responses to Questions

**Comments to the Author**

1. If the authors have adequately addressed your comments raised in a previous round of review and you feel that this manuscript is now acceptable for publication, you may indicate that here to bypass the “Comments to the Author” section, enter your conflict of interest statement in the “Confidential to Editor” section, and submit your "Accept" recommendation.

Reviewer #1: All comments have been addressed

2. Is the manuscript technically sound, and do the data support the conclusions?

Reviewer #1: Yes

3. Has the statistical analysis been performed appropriately and rigorously? 

Reviewer #1: Yes

4. Have the authors made all data underlying the findings in their manuscript fully available?

Reviewer #1: Yes

5. Is the manuscript presented in an intelligible fashion and written in standard English?

Reviewer #1: Yes

6. Review Comments to the Author

Reviewer #1: (No Response)

7. PLOS authors have the option to publish the peer review history of their article (what does this mean?). If published, this will include your full peer review and any attached files.

Reviewer #1: No

---

## [Editor Report · Acceptance letter]

11 Jan 2021

PONE-D-20-32202R1 

Anticipatory behaviour as an indicator of the welfare of dairy calves in different housing environments 

Dear Dr. Zobel:

I'm pleased to inform you that your manuscript has been deemed suitable for publication in PLOS ONE. Congratulations! Your manuscript is now with our production department. 

Kind regards, 

on behalf of

Prof. Dr. Arda Yildirim 

Academic Editor

PLOS ONE